# Does lack of parental involvement affect school dropout among Indian adolescents? evidence from a panel study

Ronak Paul[1], Rashmi Rashmi[2], Shobhit Srivastava[3] *

1 Department of Public Health and Mortality Studies, International Institute for Population Sciences, Mumbai, Maharashtra, India, 2 Department of Fertility Studies, International Institute for Population Sciences, Mumbai, Maharashtra, India, 3 Department of Mathematical Demography and Statistics, International Institute for Population Sciences, Mumbai, Maharashtra, India

☯ These authors contributed equally to this work.
* shobhitsrivastava889@gmail.com

**Data Availability Statement:** This study used a publicly available secondary dataset with no information that could lead to the identification of the respondents. The IHDS datasets used in our study can be downloaded from the Inter-University

## Abstract

Despite the gross enrolment ratio of Indian children, being almost 91% in grades 6–8, the equivalently soaring rates of school dropout after 8th grade remains a huge concern for the policymakers. Researches from the developed countries and some developing countries have shown the benefits of parental involvement in their children's education in terms of reduced dropout rates. However, there is a stark absence of similar evidence in the Indian context. Our study examines whether the lack of parental involvement during primary schooling of Indian children eventually results in school dropout when the children become adolescents. We used IHDS panel data of children (8–11 years) in round-I who become adolescents (15–18 years) in round-II. Bivariate, multivariable and stratified analyses were performed using logistic regression models. The findings from the multivariable models show that children, whose parents did not -participate in PTA meetings, -discuss academic progress with schoolteacher and -supervise their children's homework in round-I respectively had 1.15 (95% CI: 1.01–1.30), 1.14 (95% CI: 1.01–1.29) and 1.17 (95% CI: 1.01–1.34) times higher risk of school dropout in round-II. Further, a similar relationship was observed when hypothesized relationship by gender, type of school attended and type of community of the children were examined. Among male children, parents' non-participation in PTA meetings was associated with 1.21 (95% CI: 1.02–1.44) times greater odds of school dropout. Children from private schools also had a 2.17 (95% CI: 1.42–3.32) times greater risk of dropout if their parents did not supervise their children in homework These findings highlight the crucial role of parental involvement in their children's primary education, in terms of reduced school dropout. The findings call for programmatic interventions that create awareness and encourage parental participation in their children's schooling.

Consortium for Political and Social Research (ICPSR) data repository. The data are available at https://www.icpsr.umich.edu/web/DSDR/studies/36151/versions/V6 and https://www.icpsr.umich.edu/web/DSDR/studies/22626.

**Funding:** The author(s) received no specific funding for this work.

**Competing interests:** The authors have declared that no competing interests exist.

## Introduction

With the global commitment of Education for All, India started moving towards the goal of universal elementary education in 1992. While initiatives like the Right to Education Act, Sarva Shiksha Abhiyan, the Mid-day meal scheme and many more, resulted in rapid increment of primary school enrolments, the issue of discontinuation of schooling education had grown unacceptably in India [1]. As per National Education Policy (NEP) report, the gross enrolment ratio (GER) for grades 6–8 was 90.9%, while for grades 9–10 and 11–12 it was 79.3% and 56.5% respectively [2]. This shows the successful effort of bringing children under the formal education system through primary schooling. However, the increasing dropout rate among Indian children, especially after 8th grade, has put the long-term benefits of such gross enrolment into question. "No detention policy", that ruled out grade retention upto 8th grade, brought down the dropout rates to half during 2014–15 from the highs of 2006–07. However, the prevalent rate of school dropout among adolescents is still a major cause of concern [3]. According to NFHS-4, 3.9% of male and 3.2%, female children experience dropout due to repeated failure in school [4]. Grade repetition emerged as a stressful event for early adolescents [5, 6]. Repetitive failures not only affects the confidence but inculcates negative attitude among children which further disrupts their continuation to secondary schooling [6, 7]. Moreover, school dropouts are responsible for long-term consequences like illiteracy, unemployment, low wage, child labour, mental health issues and involvement in criminal activities [8–10].

Besides socio-economic, household and child-related characteristics, parental participation in home and school is seen as an important predictor of education and development among children in both developed and developing countries [11, 12]. Parental involvement in child schooling indicates the role of parents in guiding the children in their learning process as well as dedicating time to look after the vicissitudes of their life and career. Existing researches have used different definitions of parental involvement. Joyce Epstein came with a typology of parental involvement that includes good parenting (providing housing, nutrition and interacting with child), communication with the school, volunteering in classrooms or events, teaching at home (educational choice and help in homework), decision-making (participation in PTA) and collaborating with the community [13]. In developed countries, research had shown that children at any age are benefitted from a certain amount of parental involvement [14]. Another study found that parents who continually motivate their children for doing their best in whatever activities they like had helped improve achievement among children [15]. Additionally, one study highlighted the benefits of parental involvement, among Spanish adolescents, in terms of better academic achievement [16]. School composition and peer group was also considered to be a crucial determinant of child schooling and education after 16 years of age [11]. One study of British children had shown the negative impact of maternal deprivation on their educational attainment [17]. Taken together these studies suggest that a majority of children benefit from experiencing parental involvement during their elementary education.

Despite such well-established benefits of parental involvement in schooling across developed and developing countries, there is a lack of similar research in the Indian context. Contemporary research in developing countries talks about factors that affect educational attainment and dropout among children [18–20]. Particularly in the case of India, illiteracy among parents, poverty, the gender of the children, family size and religion are proven roadblocks to the quality and continuity of education in India [21–25]. One study had shown the association of several household characteristics with school dropout among Indian children [26]. A couple of Indian cross-sectional studies had also talked about the negative association of lack of parental involvement on the continuity and achievement of formal education among

children at the elementary level [27, 28]. Another study found parental aspirations as an important predictor of a child's schooling and achievements [29]. The evidence found from this Indian literature had also shown a differential in the association across the residence, gender and type of school facilities. However, there is a dearth of evidence showing the impact of lack of parental involvement during primary schooling on the continuity of their children's education in a later course. This gives us the point of departure for the present study. The objective of our study is to examine whether the lack of parental involvement during primary schooling of the children eventually result in detrimental outcomes, in terms of school dropout, when the children become adolescents. To fulfil this objective, we use the India Human Development Survey (IHDS) panel data for children aged 8–11 years in round-I who become adolescents aged 15–18 years in round-II. Our study hypothesizes that there is no relationship of lack of parental involvement in round-I with the school dropout status of adolescents in round-II. Further, we examine whether this hypothesized relationship varies across different subsets of the Indian population.

## Methods

### Data source

This study used round-I and round-II of the India Human Development Survey (IHDS), conducted by the National Council of Applied Economic Research (NCAER) in collaboration with the University of Maryland, USA. IHDS round-I is a nationally representative survey that collected information from 41,554 households across all states and union territories of India except Andaman & Nicobar Islands and Lakshadweep, during 2004–05 [30]. IHDS round-II carried out during 2011–12, collected information from 42,152 households with geographical coverage similar to round-I [31]. IHDS round-II re-interviewed 83% of the households from round-I. IHDS adopted a stratified random sampling survey design and informed consent were obtained from all the interviewee. Further details regarding survey description, sampling design and data quality can be found elsewhere [32–34]. Additional information on informed consent is available from the survey questionnaires available from the IHDS website [35, 36].

Our study utilized the panel data for 9840 children aged 8–11 years in round-I who became 15–18 years old during round-II. There were 17,061 children aged 8–11 years in round-I among whom 104 died, 3,454 migrated and 3,663 children were untraceable during round-II. Further, we excluded the data for 122 children who had missing information regarding their school dropout status in round-II. Therefore, for investigating the relationship between parental involvement in round-I with school dropout in round-II, the analytical sample size is 9718 adolescents. Among 9,718 children, 7,445 (77%) children were enrolled for schooling education in both rounds. However, 2,273 (23%) children who were enrolled in the round–I had experienced school dropout in round-II.

### Ethics statement

This study used a publicly available secondary dataset with no information that could lead to the identification of the respondents. The IHDS datasets used in our study can be downloaded from the Inter-University Consortium for Political and Social Research (ICPSR) data repository [35, 36].

### Outcome variables

The outcome variables of this study are a binary indicator of whether a student dropped out of school between round–I and–II when they become aged 15–18 years (adolescents) during

round-II. School dropout statuses of students were obtained from the binary indicators of their school enrolment status collected during both rounds of IHDS. We have included only those children who were enrolled in a school during round-I. Among them, children who were enrolled in school during round-I but were not enrolled during round-II were categorized as "yes" (school dropout) and those who were enrolled during both rounds were categorized as "no".

## Explanatory variables

The three binary indicators of lack of parental involvement are the explanatory variables in this study. These three indicators are–whether the parents participate in parent-teacher association (PTA) meetings; parents discussed the academic progress of the students with their schoolteacher; and, parents supervise the students while doing homework. These three variables were measured for children aged 8–11 during round-I and have been categorized into "yes" and "no".

The variables for parental participation in PTA meetings, and, whether parents discussed the academic progress of their children with the schoolteacher within a year, was constructed from the similar question that IHDS asked from parents of children aged 8–11 in round-I. Further, during round-I information was collected regarding whether the mother, any adult men, any adult women or other children of the household supervises the children while doing homework. If anyone supervised the students while doing homework, then they were coded as "yes" and otherwise were coded into "no".

## Control variables

Existing studies show that several factors other than lack of parental involvement also influence the school dropout of students. We controlled for the confounding effect of these relevant factors in our study, conditional to their availability in IHDS datasets. The confounding factors related to the student and their school are–age of the student in years, the gender of the student (male, female), type of student (better than average, average and below), type of school attended by the student (public school, private school), the student takes private tuition (no, yes). We also controlled for parent-related characteristics–mother's level of education (no formal schooling, less than 5 years of schooling, 6–10 years of schooling, more than 10 years of schooling), mother's working status (not working, working), father's level of education (no formal schooling, less than 5 years of schooling, 6–10 years of schooling, more than 10 years of schooling), father's working status (not working, working). Further, the socio-economic characteristics of the student households were also included–household wealth quintile (richest, rich, middle, poor, poorest), household below poverty line (BPL) status (non-poor, poor), caste of the household (scheduled tribes (ST), scheduled castes (SC), other backward classes (OBC), others), religion of the household (Hindu, Muslim, others), type of community the student belongs (rural, urban), country region a student comes from–(northern, north-eastern, central, eastern, western, southern). All these factors were measured for the panel of children aged 8–11 during round-I.

Additionally, we included the binary variable of whether the students had repeated a grade between round-I and round-II. We constructed this variable from the two binary variables of whether the students had ever repeated a grade during round-I and round-II respectively. Those students, who had not repeated any grade in both rounds were categorized as "No repeat" and who had not repeated a grade in round-I but had repeated grade in round-II were categorized as "Repeat".

Household wealth quintile was measured in round-I using principal component analysis [37]. Wealth scores for each household were generated using the information on household asset ownership, livestock ownership, building material used in household, household water source, household sanitation facility and the number of rooms. Based on the wealth score the households were classified into five categories (poorest, poor, middle, rich, richest) such that the households with the lowest 20 percentile score belonged to the "poorest" category, households with the next low 20 percentile score belonged to the "poor" category and so forth.

The country regions during round-I were formed by including the erstwhile 33 states and union territories of India into six categories. The northern region includes Chandigarh, Delhi, Haryana Himachal Pradesh, erstwhile Jammu & Kashmir, Punjab, Uttaranchal and Rajasthan. The north-eastern region includes Assam, Arunachal Pradesh, Manipur, Meghalaya, Mizoram, Nagaland, Tripura and Sikkim. The central region consists of Madhya Pradesh and Chhattisgarh. The eastern zone consists of Bihar, Jharkhand, Odisha and West Bengal. The western region comprises Dadra & Nagar Haveli, Daman & Diu, Goa, Gujarat and Maharashtra. The southern region comprises erstwhile Andhra Pradesh, Karnataka, Kerala, Tamil Nadu and Pondicherry.

## Statistical methods

We performed bivariate and multivariable analysis using logistic regression models to achieve the study objectives. Owing to the binary nature of the outcome variable, we performed bivariate analysis using the chi-square test for association. Equivalently, we undertook multivariate analysis by estimating multivariable logistic regression models. In the multivariable models, the association between parental involvement in round-I and school dropout in round-II was shown using odds ratios. Odds ratio gives the odds of school dropout of adolescents, from one category of an explanatory variable in comparison to the reference category of that explanatory variable after controlling for the effect of other confounding factors, relative to those adolescents who did not experience school dropout [38].

Further, we performed a stratified multivariable analysis to check for the differential impact, of lack of parental involvement in round-I on the school dropout of adolescents in round-II, by their gender, type of school attended and type of community they belong to. The first set of stratified analysis models involved estimating separate multivariable logistic regression models for subsamples of male and female children. In the second and third sets of regression models, we divided the full sample into subsamples of public-private school and rural-urban children respectively.

We checked for multicollinearity in the multiple variable regression models and the mean values of variance inflation factor (VIF) for each of the models were less than 1.3. Therefore, multicollinearity does not affect our estimated models [38]. We also checked for possible interaction effects between the explanatory variables used in our study [39]. We found evidence of interaction effect between–grade repetition and country region, wealth quintile and country region, religion and country region. However, we did not find suitable explanations in existing literature for these observed interaction effects, in the Indian context, and therefore did not include them in our statistical models. Our study results are un-weighted, as the use of panel data requires the application of panel weights. However, IHDS does not provide separate panel weights for analysis. All the statistical estimations were done using the STATA software version 13.0 [40].

## Results

### Sample description

*Table 1* shows the absolute and percentage distribution of children aged 8–11 by relevant parental, demographic and socio-economic characteristics during round-I. We found that

among the panel of children 55%, 29% and 13% of the children have parents who did not–attend PTA meetings,–discuss academic progress with the schoolteacher and–supervise their children while doing homework respectively. When we come to demographic characteristics, 53% of children were male and 75% attended public school. Furthermore, the father and mother of 55% and 26% of children have had no formal schooling respectively. We also observe that 29% of children come from households below the poverty line and 7% and 23% of children belonged to the ST and SC category respectively. Moreover, 79% of children belonged to a Hindu household and 72% come from a rural community. Coming to geographic distribution, a majority (38%) of the children come from the northern region followed by 18% and 15% coming from the southern and eastern regions of India respectively. We observed that the percentage difference of children by demographic, socio-economic and geographic characteristics was similar between the cross-sectional and panel datasets. Only percentage distribution by age of the children (in years) varied by more than 2% between the two datasets.

## Bivariate analysis

Section 1 of *Table 2* shows the bivariate association between the lack of parental involvement in round-I with the dropout status of adolescents in round-II. Among 9,718 children, 2,273 had experienced school dropout during adolescence. The bivariate results show that 29% of children whose parents did not participate in PTA meetings during round-I had experienced school dropout in round-II. Further, we find that parents who do not discuss the academic progress of their children with the school teacher and do not supervise the homework, those children had a 32% and 34% chance of school dropout in round-II respectively. 36% of children who had repeated their grade between two rounds experienced school dropout in round-II. Nearly 25% of children who were average and below-average students in round-I had dropout from school in round-II. Most of the children (28%) who were from public schools in round-I, had discontinued their schooling in round-II. 32% and 37% of children whose mother and father had no formal schooling in round-I, respectively, experience dropout in round-II. Interestingly, dropout was common among 25% of children whose fathers were working in round-I. Children belonging to Scheduled Tribes and from rural community experienced 36% and 25% dropout in round-II.

## Multivariable analysis

After controlling different characteristics, multivariable logistic regression in section 2 of *Table 2* shows the association of lack of parental involvement in round-I with the school dropout status of adolescents in round-II. The multivariable analysis shows that if the parents did not participate in PTA meetings during round-I then their children had 1.15 (95% CI: 1.01–1.30) times higher chances of school dropout in round-II. Moreover, the children whose parents did not discuss their academic progress with the schoolteacher in round-I had 1.14 (95% CI: 1.01–1.29) times higher odds of school dropout in round-II. Further, we observe that non-supervision of school homework by parents during round-I is associated with a 1.17 (95% CI: 1.01–1.34) times higher risk of school dropout among their children in round-II. Additionally, we observe that children studying in public schools during round-I had 1.70 (95% CI: 1.44–2.01) times higher odds of school dropout during round-II compared to children studying in private schools. Moreover, children of mothers who had more than 10 years of schooling had 0.24 (95% CI: 0.14–0.42) times lower odds of school dropout compared to those children whose mothers had no formal schooling. Similarly, if the fathers had more than 10 years of formal education then their children had 0.36 (95% CI: 0.27–0.48) times lower odds of school dropout in round-II respectively. Further, children from households belonging to the poorest

**Table 1. Absolute and percentage distribution of children by parental involvement variables and other relevant demographic and socio-economic characteristics across the cross-sectional and panel datasets for children aged 8–11 years in round-I.**

| Characteristics in round-I | Adolescents aged 8–11 years in round-I | | | | Absolute difference |
|---|---|---|---|---|---|
| | Cross-sectional dataset | | Panel dataset | | |
| | N | % | N | % | % |
| **Parents participate in PTA meetings** | | | | | |
| No | 9,588 | 56.2 | 5,315 | 54.7 | 1.5 |
| Yes | 7,473 | 43.8 | 4,403 | 45.3 | 1.5 |
| **Parents discussed academic progress with teacher** | | | | | |
| No | 4,873 | 28.6 | 2,809 | 28.9 | 0.3 |
| Yes | 12,188 | 71.4 | 6,909 | 71.1 | 0.3 |
| **Parents supervises while doing homework** | | | | | |
| No | 2,507 | 14.7 | 1,257 | 12.9 | 1.8 |
| Yes | 14,554 | 85.3 | 8,461 | 87.1 | 1.8 |
| **Age of the student (in years)** | | | | | |
| 8 | 4,311 | 25.3 | 2,115 | 21.8 | 3.5 |
| 9 | 3,714 | 21.8 | 2,293 | 23.6 | 1.8 |
| 10 | 5,596 | 32.8 | 3,510 | 36.1 | 3.3 |
| 11 | 3,440 | 20.2 | 1,800 | 18.5 | 1.7 |
| **Gender of the student** | | | | | |
| Male | 8,940 | 52.4 | 5,188 | 53.4 | 1.0 |
| Female | 8,121 | 47.6 | 4,530 | 46.6 | 1.0 |
| **Type of student** | | | | | |
| Better than average | 2,006 | 11.8 | 1,192 | 12.3 | 0.5 |
| Average and below | 15,055 | 88.2 | 8,526 | 87.7 | 0.5 |
| **Type of school attended by students** | | | | | |
| Public School | 12,894 | 75.6 | 7,301 | 75.1 | 0.5 |
| Private School | 4,167 | 24.4 | 2,417 | 24.9 | 0.5 |
| **Student takes private tuition** | | | | | |
| No | 14,198 | 83.2 | 8,013 | 82.5 | 0.7 |
| Yes | 2,863 | 16.8 | 1,705 | 17.5 | 0.7 |
| **Mother's level of education** | | | | | |
| No formal schooling | 9,668 | 56.7 | 5,330 | 54.8 | 1.9 |
| Less than 5 years of schooling | 2,595 | 15.2 | 1,621 | 16.7 | 1.5 |
| 6–10 years of schooling | 3,717 | 21.8 | 2,172 | 22.4 | 0.6 |
| More than 10 years of schooling | 1,081 | 6.3 | 595 | 6.1 | 0.2 |
| **Mother's working status** | | | | | |
| Not working | 12,684 | 74.3 | 7,231 | 74.4 | 0.1 |
| Working | 4,377 | 25.7 | 2,487 | 25.6 | 0.1 |
| **Father's level of education** | | | | | |
| No formal schooling | 4,654 | 27.3 | 2,524 | 26.0 | 1.3 |
| Less than 5 years of schooling | 2,913 | 17.1 | 1,715 | 17.6 | 0.5 |
| 6–10 years of schooling | 7,137 | 41.8 | 4,183 | 43.0 | 1.2 |
| More than 10 years of schooling | 2,357 | 13.8 | 1,296 | 13.3 | 0.5 |
| **Father's working status** | | | | | |
| Not working | 5,189 | 30.4 | 3,095 | 31.8 | 1.4 |
| Working | 11,872 | 69.6 | 6,623 | 68.2 | 1.4 |
| **Household wealth quintile** | | | | | |
| Richest | 3,285 | 19.3 | 1,905 | 19.6 | 0.3 |

*(Continued)*

**Table 1.** (Continued)

| Characteristics in round-I | Adolescents aged 8–11 years in round-I | | | | Absolute difference |
|---|---|---|---|---|---|
| | Cross-sectional dataset | | Panel dataset | | |
| | N | % | N | % | % |
| Rich | 3,593 | 21.1 | 2,089 | 21.5 | 0.4 |
| Middle | 3,437 | 20.1 | 2,052 | 21.1 | 1.0 |
| Poor | 3,404 | 20.0 | 1,904 | 19.6 | 0.4 |
| Poorest | 3,342 | 19.6 | 1,768 | 18.2 | 1.4 |
| Household BPL status[a] | | | | | |
| Not poor | 12,146 | 71.2 | 6,951 | 71.5 | 0.3 |
| Poor | 4,915 | 28.8 | 2,767 | 28.5 | 0.3 |
| Caste of household | | | | | |
| Scheduled Tribes | 1,333 | 7.8 | 687 | 7.1 | 0.7 |
| Scheduled Castes | 3,729 | 21.9 | 2,212 | 22.8 | 0.9 |
| Other Backward Classes | 6,889 | 40.4 | 3,884 | 40.0 | 0.4 |
| Others | 5,110 | 30.0 | 2,935 | 30.2 | 0.2 |
| Religion of household | | | | | |
| Hindu | 13,353 | 78.3 | 7,716 | 79.4 | 1.1 |
| Muslim | 2,532 | 14.8 | 1,320 | 13.6 | 1.2 |
| Others | 1,176 | 6.9 | 682 | 7.0 | 0.1 |
| Type of community | | | | | |
| Rural | 12,040 | 70.6 | 6,957 | 71.6 | 1.0 |
| Urban | 5,021 | 29.4 | 2,761 | 28.4 | 1.0 |
| Country region | | | | | |
| Northern | 6,339 | 37.2 | 3,731 | 38.4 | 1.2 |
| North Eastern | 615 | 3.6 | 256 | 2.6 | 1.0 |
| Central | 1,888 | 11.1 | 1,084 | 11.2 | 0.1 |
| Eastern | 2,845 | 16.7 | 1,496 | 15.4 | 1.3 |
| Western | 2,210 | 13.0 | 1,427 | 14.7 | 1.7 |
| Southern | 3,164 | 18.5 | 1,724 | 17.7 | 0.8 |
| **Overall** | **17,061** | **100** | **9,718** | **100** | **0** |

Note–

(a) BPL: Below Poverty Line

(b) N: Sample

(c) %: Percentage.

wealth quintile had 3.16 (95% CI: 2.41–4.15) times greater chances of dropout in comparison to the children from the richest quintile households. Furthermore, we find that children in the urban community had 1.43 (95% CI: 1.24–1.66) times higher odds of school dropout compared to their rural community counterparts.

## Stratified analysis by gender, type of school attended and type of community

From *Table 3* we observe that the rates of the lack of parental involvement vary by gender, type of school attended and type of community. There is heterogeneity in the relationship between parental involvement and school dropout among adolescents. In comparison to female adolescent's, lesser dropout is experienced among male counterparts when their parents participate

**Table 2. Bivariate and multivariate association of parental involvement and other relevant demographic and socio-economic characteristics in round-I with the school dropout status of adolescents in round-II.**

| Characteristics in round-I | Adolescents aged 15–18 years in round-II | | | | | |
| --- | --- | --- | --- | --- | --- | --- |
| | (1) | | | | (2) | |
| | Total | School dropout | | Chi-square test | School dropout | |
| | N | N | % | | Odds ratio | 95% CI |
| **Parents participate in PTA meetings** | | | | | | |
| No | 5,315 | 1,528 | 28.7 | * | Ref. | |
| Yes | 4,403 | 745 | 16.9 | | 1.15* | (1.01–1.30) |
| **Parents discussed academic progress with teacher** | | | | | | |
| No | 2,809 | 884 | 31.5 | * | Ref. | |
| Yes | 6,909 | 1,389 | 20.1 | | 1.14* | (1.01–1.29) |
| **Parents supervises while doing homework** | | | | | | |
| No | 1,257 | 429 | 34.1 | * | Ref. | |
| Yes | 8,461 | 1,844 | 21.8 | | 1.17* | (1.01–1.34) |
| **Age of the student (in years)** | | | | | | |
| 8 | 2,115 | 371 | 17.5 | * | Ref. | |
| 9 | 2,293 | 461 | 20.1 | | 1.34* | (1.13–1.57) |
| 10 | 3,510 | 926 | 26.4 | | 1.84* | (1.59–2.13) |
| 11 | 1,800 | 515 | 28.6 | | 2.28* | (1.93–2.70) |
| **Gender of the student** | | | | | | |
| Male | 5,188 | 1,196 | 23.1 | # | Ref. | |
| Female | 4,530 | 1,077 | 23.8 | | 1.11 | (1.00–1.23) |
| **Type of student** | | | | | | |
| Better than average | 1,192 | 160 | 13.4 | * | Ref. | |
| Average and below | 8,526 | 2,113 | 24.8 | | 1.34* | (1.11–1.62) |
| **Ever repeated grade[(d)]** | | | | | | |
| No repeat | 8,061 | 1,677 | 20.8 | * | Ref. | |
| Repeat | 1,657 | 596 | 36.0 | | 1.85* | (1.63–2.10) |
| **Type of school attended by students** | | | | | | |
| Public School | 7,301 | 2,043 | 28.0 | * | Ref. | |
| Private School | 2,417 | 230 | 9.5 | | 1.70* | (1.44–2.01) |
| **Student takes private tuition** | | | | | | |
| No | 8,013 | 2,058 | 25.7 | * | Ref. | |
| Yes | 1,705 | 215 | 12.6 | | 1.49* | (1.25–1.78) |
| **Mother's level of education** | | | | | | |
| No formal schooling | 5,330 | 1,727 | 32.4 | * | Ref. | |
| Less than 5 years of schooling | 1,621 | 334 | 20.6 | | 0.68* | (0.59–0.79) |
| 6–10 years of schooling | 2,172 | 197 | 9.1 | | 0.44* | (0.37–0.53) |
| More than 10 years of schooling | 595 | 15 | 2.5 | | 0.24* | (0.14–0.42) |
| **Mother's working status** | | | | | | |
| Not working | 7,231 | 1,446 | 20.0 | * | Ref. | |
| Working | 2,487 | 827 | 33.3 | | 1.07 | (0.94–1.22) |
| **Father's level of education** | | | | | | |
| No formal schooling | 2,524 | 940 | 37.2 | * | Ref. | |
| Less than 5 years of schooling | 1,715 | 558 | 32.5 | | 1.03 | (0.89–1.19) |
| 6–10 years of schooling | 4,183 | 704 | 16.8 | | 0.65* | (0.57–0.75) |
| More than 10 years of schooling | 1,296 | 71 | 5.5 | | 0.36* | (0.27–0.48) |
| **Father's working status** | | | | | | |

*(Continued)*

**Table 2.** (Continued)

| Characteristics in round-I | Adolescents aged 15–18 years in round-II | | | | | |
|---|---|---|---|---|---|---|
| | (1) | | | | (2) | |
| | Total | School dropout | | Chi-square test | School dropout | |
| | N | N | % | | Odds ratio | 95% CI |
| Not working | 3,095 | 604 | 19.5 | * | Ref. | |
| Working | 6,623 | 1,669 | 25.2 | | 0.94 | (0.83–1.07) |
| **Household wealth quintile** | | | | | | |
| Richest | 1,905 | 118 | 6.2 | * | Ref. | |
| Rich | 2,089 | 339 | 16.2 | | 1.54* | (1.21–1.95) |
| Middle | 2,052 | 531 | 25.9 | | 2.28* | (1.79–2.91) |
| Poor | 1,904 | 574 | 30.1 | | 2.35* | (1.82–3.04) |
| Poorest | 1,768 | 711 | 40.2 | | 3.16* | (2.41–4.15) |
| **Household BPL status[(e)]** | | | | | | |
| Not poor | 6,951 | 1,285 | 18.5 | * | Ref. | |
| Poor | 2,767 | 988 | 35.7 | | 1.26* | (1.12–1.42) |
| **Caste of household** | | | | | | |
| Scheduled Tribes | 687 | 249 | 36.2 | * | Ref. | |
| Scheduled Castes | 2,212 | 590 | 26.7 | | 1.03 | (0.83–1.26) |
| Other Backward Classes | 3,884 | 947 | 24.4 | | 1.00 | (0.82–1.22) |
| Others | 2,935 | 487 | 16.6 | | 0.81 | (0.65–1.02) |
| **Religion of household** | | | | | | |
| Hindu | 7,716 | 1,699 | 22.0 | * | Ref. | |
| Muslim | 1,320 | 442 | 33.5 | | 1.98* | (1.70–2.32) |
| Others | 682 | 132 | 19.4 | | 1.21 | (0.96–1.53) |
| **Type of community** | | | | | | |
| Rural | 6,957 | 1,767 | 25.4 | * | Ref. | |
| Urban | 2,761 | 506 | 18.3 | | 1.43* | (1.24–1.66) |
| **Country region** | | | | | | |
| Northern | 3,731 | 744 | 19.9 | * | Ref. | |
| North Eastern | 256 | 58 | 22.7 | | 1.42 | (1.00–2.01) |
| Central | 1,084 | 328 | 30.3 | | 1.07 | (0.89–1.29) |
| Eastern | 1,496 | 371 | 24.8 | | 0.97 | (0.82–1.16) |
| Western | 1,427 | 389 | 27.3 | | 2.04* | (1.71–2.42) |
| Southern | 1,724 | 383 | 22.2 | | 1.23* | (1.04–1.46) |
| **Overall** | **9,718** | **2,273** | **23.4** | | **9,718** | |

Note–(1) Bivariate association shown using Chi-square test for association; (2) Multivariate association shown using odds ratios from multivariable logistic regression;
(a) Ref.: reference category; (b) Statistical significance denoted by asterisks: * p-value<0.05 (significant), # p-value>0.05; (c) 95% Confidence interval is given in brackets
(d) Shows whether a student had ever repeated grade between round-I and round-II
(e) BPL: Below Poverty Line.

in PTA meeting during primary schooling. Private school children were found to be more advantageous when any form of parental involvement is seen during their primary education.

Therefore, we ran separate regression models for male and female children, children attending public and private schools and children from rural and urban communities respectively and the results for the same are shown in *Table 4*. Table 4 shows the regression results for male and female children. Among male children, parents' non-participation in PTA meetings was associated with 1.21 (95% CI: 1.02–1.44) times greater odds of school dropout.

**Table 3. Absolute and percentage distribution of children by the parental involvement variables by gender, type of school attended and type of community of the students during round-I.**

| Characteristics | Total population | (1) | | (2) | | (3) | |
|---|---|---|---|---|---|---|---|
| | | Yes | Yes | Yes | Yes | Yes | Yes |
| | N | N | % | N | % | N | % |
| **Gender of the student** | | | | | | | |
| Male | 5,188 | 2,308 | 44.5 | 3,644 | 70.2 | 4,494 | 86.6 |
| Female | 4,530 | 2,095 | 46.2 | 3,265 | 72.1 | 3,967 | 87.6 |
| **Type of school attended by students** | | | | | | | |
| Public School | 7,301 | 2,887 | 39.5 | 4,890 | 67.0 | 6,252 | 85.6 |
| Private School | 2,417 | 1,516 | 62.7 | 2,019 | 83.5 | 2,209 | 91.4 |
| **Type of community** | | | | | | | |
| Rural | 6,957 | 2,827 | 40.6 | 4,734 | 68.0 | 5,965 | 85.7 |
| Urban | 2,761 | 1,576 | 57.1 | 2,175 | 78.8 | 2,496 | 90.4 |
| **Overall** | **9,718** | **4,403** | **45.3** | **6,909** | **71.1** | **8,461** | **87.1** |

Note–(1) Parents participate in PTA meetings; (2) Parents discussed academic progress with the teacher; (3) Parents supervises while doing homework; (a) N: Sample; (b): %: Percentage.

Comparatively, in female children lack of parental participation in the form of academic discussion with the teacher was positively associated with the risk of school dropout. The results for children from public and private schools are shown in Table 4. Non-participation in PTA meetings and non-discussion of academic progress with schoolteacher during round-I is associated with greater chances of school dropout among students of public school in round-II. Moreover, children from private schools also had a 2.17 (95% CI: 1.42–3.32) times greater risk of dropout if their parents did not supervise their children in homework. Interesting results appear when we look at the association of the lack of parental involvement with the dropout status of children from a rural and urban community in Table 4. While non-supervision of homework by parents has a statistically significant positive association with school dropout among urban children, non-participation in PTA meetings and non-discussion of academic progress was associated with a greater risk of school dropout among rural children.

## Discussion

The present study examined the effect of parental participation in their children's primary school education on the educational outcomes of secondary school (i.e., when they reach their adolescence phase) in terms of school dropout. Based on IHDS panel data, this study provides evidence that Indian children whose parents did not indulge in their primary stage learning process; were more likely to be affected by negative educational outcomes at their adolescent phase. School dropout was common among those adolescents whose parents had not participated in PTA meetings, not discussed academic progress with the teacher and not supervised their homework during primary schooling. These findings were consistent with one existing study which showed that dropout was high among American families in which parents were less involved in the education of children [41]. Similar to our study, another study on Icelandic youths had also shown the importance of parent-child relationship quality for reducing the risk of school dropout [42]. Similar to our findings, another study had also shown that parent's active communication with teachers and family involvement in school-related activities usually lower the chances for dropouts in lower secondary schooling [43].

**Table 4. Adjusted odds ratios from logistic regression models showing the association between parental involvement in round-I with the school dropout and grade status in round-II by gender, type of school attended and type of community of the students.**

| Characteristics in round-I | School dropout among adolescents in round-II | | | |
|---|---|---|---|---|
| | Male | | Female | |
| | Odds ratio | 95% CI | Odds ratio | 95% CI |
| **Parents participate in PTA meetings** | | | | |
| Yes | Ref. | | Ref. | |
| No | 1.21* | (1.02–1.44) | 1.06 | (0.89–1.27) |
| **Parents discussed academic progress with teacher** | | | | |
| Yes | Ref. | | Ref. | |
| No | 1.07 | (0.91–1.27) | 1.23* | (1.03–1.48) |
| **Parents supervises while doing homework** | | | | |
| Yes | Ref. | | Ref. | |
| No | 1.19 | (0.98–1.44) | 1.13 | (0.91–1.39) |
| **Analytical sample size** | 5,188 | | 4,530 | |
| | Public school | | Private school | |
| | Odds ratio | 95% CI | Odds ratio | 95% CI |
| **Parents participate in PTA meetings** | | | | |
| Yes | Ref. | | Ref. | |
| No | 1.15* | (1.01–1.31) | 1.14 | (0.79–1.63) |
| **Parents discussed academic progress with teacher** | | | | |
| Yes | Ref. | | Ref. | |
| No | 1.14* | (1.00–1.30) | 1.07 | (0.71–1.61) |
| **Parents supervises while doing homework** | | | | |
| Yes | Ref. | | Ref. | |
| No | 1.08 | (0.93–1.26) | 2.17* | (1.42–3.32) |
| **Analytical sample size** | 7,301 | | 2,417 | |
| | Rural | | Urban | |
| | Odds ratio | 95% CI | Odds ratio | 95% CI |
| **Parents participate in PTA meetings** | | | | |
| Yes | Ref. | | Ref. | |
| No | 1.51* | (1.16–1.96) | 1.06 | (0.92–1.23) |
| **Parents discussed academic progress with teacher** | | | | |
| Yes | Ref. | | Ref. | |
| No | 1.37* | (1.04–1.81) | 1.11 | (0.96–1.27) |
| **Parents supervises while doing homework** | | | | |
| Yes | Ref. | | Ref. | |
| No | 1.03 | (0.74–1.45) | 1.18* | (1.00–1.38) |
| **Analytical sample size** | 2,761 | | 6,957 | |

Note–(a) Ref. denotes reference category; (b) Statistical significance denoted by asterisks: * p-value<0.05; (c) 95% Confidence interval is given in brackets; (d) All the models controlled for the effect of all the control variables but their results have not been shown in the table.

The present study found that school dropout was higher among adolescents with average or below class performance. The results were consistent with previous research works where it was argued that high dropouts were a result of persistently low performing students being rolled out of their school, as those students were likely to hamper down the overall performance statistics of their school [44]. Besides, grade repetition was one of the risk factors for higher school dropout among adolescents in our study. The findings were parallel with the existing findings that poor children are at risk to enter school at later ages, repeat grades and

then more often leave school early [10]. Other reasons may be that grade retention makes students overage for a grade, which in turn causes them to drop out of school [45].

Dropouts were more common among the public-school children in this study. These results were consistent with an existing study that showed that dropout was higher in public schools due to poor performances of children and a huge shortage of teachers which creates lesser motivation among parents for sending their children to schools [46]. Moreover, one Indian study showed that infrastructure and schooling cost significantly varies by type of ownership of schools. Children of privately run schools with better infrastructure and higher schooling cost outperforms the children going to publicly run schools [47]. Further, in the present study lower chance of school dropout was observed among adolescents whose parents had higher educational attainment. This evidence was again consistent with one existing study where illiterate parents show less encouragement towards their children's education [48]. Moreover, similar to existing studies our study also found that high parental income and better socio-economic status paved the way for a reduction in dropout status as children coming from such background were provided better resources including access to better quality schools, private tuitions and more support for learning within the home [49–51]. Further, consistent with one study our study also found that Indian children belonging to the SC and ST category show higher dropout rates than those of other categories [52].

Literature from the developed and a few developing countries had consistently shown the importance of parental involvement in a child's education [53]. However, this study had tried to strengthen the literature in developing countries and explored such association in the context of Indian adolescents. Few studies had brought forward the role of parents in universalizing and continuation of elementary education in India. However, with the growing rates of dropout after the eighth grade in India, there is a need to understand how the parental factor is affecting the children at later ages. The panel nature of IHDS data helps us to understand such association and strengthens our results. Moreover, extant research papers based on cross-sectional studies were unable to capture the long-term consequence of parental involvement in their children's education, a research gap that our study fills up. To the best of the authors' knowledge, this study is the first to present the association of parental involvement in primary schooling on the educational outcome of children in the adolescence period in India. Furthermore, a similar relationship highlighting the detrimental impact of lack of parental involvement was observed across the relevant subsets (by gender, type of school, poverty status and type of community) of the whole population. This shows that the findings are not sensitive to unobserved bias. Moreover, this study takes advantage of nationally representative data, which helps us to generalize our results for Indian children than those of the existing, state or region-specific studies, in the Indian context.

However, the study has shortcomings too. Firstly, there is a need to control for school-related characteristics like proper water and sanitation facility in schools, availability of teachers and learning resources along with a better environment, as these affect the dropout status of children. Secondly, the study results are un-weighted due to the non-availability of panel weights. Also, we were not able to capture the effect of the Right to Education act entitled for under 15 years age children on their dropout status at later ages due to unavailability of data in the survey. Moreover, factors like the number of parent-teacher meetings and the duration of time for such involvement are crucial for examining the association of meaningful parental involvement with school dropout. However, the unavailability of such information in the IHDS does not allow us to include these variables. However, besides these limitations, the study provided crucial findings that are of utmost importance in the field of dropout status of adolescents.

## Conclusion

This study provides conclusive evidence of the detrimental effect of the lack of parental involvement on their children's academic progress. Policymakers from India have mostly focused on socio-economic, household and school characteristics while making policy for children's education. However, the effects of parental involvement in their children's education are often overlooked. India is on way to adopt a new National Education Policy [2] to modernize the existing Indian education system. The present study highlights the importance, for policymakers, of encouraging meaningful parental involvement in the students' elementary school journey. A structured implementation of policies that would help in holding parent-teacher meets, activities for the family as a part of homework and involvement of parents during child education are required to create a healthy environment among children-parents-teachers. This would further help in reducing incidents of school dropout among adolescents, which is a requirement highlighted in the National Education Policy. Besides this one Indian study had shown that 70% of total students are present in government primary schools which increases the importance of reducing the gap between public and private run schools [47]. The study rightly suggested the need of strengthening the community level participation by forming a village education committee and monitoring the teacher's activities along with infrastructure planning. Stating this study as the foundation, the present study deepens the need of inculcating different measures in public and private schools to reduce the proportion of discontinuation from schools. Moving beyond this, the present study recommend the sensitization of parents through teachers, schools and community to make them aware of their ever-important role in the learning process of their children.

## Author Contributions

**Conceptualization:** Ronak Paul, Rashmi Rashmi, Shobhit Srivastava.

**Data curation:** Rashmi Rashmi.

**Formal analysis:** Ronak Paul.

**Investigation:** Ronak Paul, Rashmi Rashmi.

**Methodology:** Rashmi Rashmi.

**Resources:** Ronak Paul.

**Supervision:** Shobhit Srivastava.

**Validation:** Ronak Paul.

**Writing – original draft:** Ronak Paul, Rashmi Rashmi, Shobhit Srivastava.

**Writing – review & editing:** Ronak Paul, Rashmi Rashmi, Shobhit Srivastava.

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
