## [Decision Letter · Decision Letter 0]

16 Feb 2021

PONE-D-20-34729

Does Lack of Parental Involvement Affect School Dropout among Indian Adolescents? Evidence from a Panel Study

PLOS ONE

Dear Dr. Srivastava,

Thank you for submitting your manuscript to PLOS ONE. After careful consideration, we feel that it has merit but does not fully meet PLOS ONE’s publication criteria as it currently stands. Therefore, we invite you to submit a revised version of the manuscript that addresses the points raised during the review process.

Revised and resubmit according to reviewer comments. Additionally, also look at heterogenous effects of the parental involvement by their educational standards and economic position. Parental involvement always not be in positive direction. Teacher and Parent meetings practice is mostly in private schools, thus there might be sample selection bias in the analyses. Kindly, examine this point as well. I will give my detailed comments once your respond to the reviewer comments. 

We look forward to receiving your revised manuscript.

Kind regards,

Srinivas Goli, Ph.D.

Academic Editor

PLOS ONE

Journal Requirements:

Additional Editor Comments:

Considering the reviewer comments, I am going with a decision of major revision. Either your respond to reviewer comments or revise according to their suggestions. We would like to see a revised version of the paper.

Reviewers' comments:

Reviewer's Responses to Questions

**Comments to the Author**

1. Is the manuscript technically sound, and do the data support the conclusions?

Reviewer #1: Yes

Reviewer #2: Yes

Reviewer #3: Partly

2. Has the statistical analysis been performed appropriately and rigorously? 

Reviewer #1: Yes

Reviewer #2: Yes

Reviewer #3: Yes

3. Have the authors made all data underlying the findings in their manuscript fully available?

Reviewer #1: Yes

Reviewer #2: Yes

Reviewer #3: Yes

4. Is the manuscript presented in an intelligible fashion and written in standard English?

Reviewer #1: Yes

Reviewer #2: Yes

Reviewer #3: Yes

5. Review Comments to the Author

Reviewer #1: The study fills a gap in the current knowledge space. It will add to the existing database and inform policy makers. However, in order to make it more robust, the following are suggested:

There could be certain concerns about the cohort size, as the adolescents were only about 50% of the original children covered. Although that seems to have been addressed, some more explanation of whether that skews the results in any manner can be made.

Another issue worth flagging is that in India, a child completes class ten by the age of 16 years, or 17 at most. Thus, there is a natural drop out as it were, around that age. If a larger number of children in the cohort are grouped towards the 17-18 years band, they would in any case drop out; either for work, or lack of higher secondary schools or colleges nearby. This could skew the results, and needs to be addressed. Besides, distance is a major factor in parents not sending their children for higher education. It will add to the study if this can be addressed.

The Right to Education Act in India addresses the entitlements of children up to 14 years. The authors could analyze whether this is a factor that comes into play in dropouts, as many benefits associated with the Act are not available after that age.

The study has brought out the need for parental involvement in furthering the educational attainment of their children. Towards this end, some policy level suggestions can be incorporated. These could be structured parent teacher meetings, some activities for the family as a part of the “home work” etc. Even though the dropout rates from public schools are higher as brought out by the study, yet these recommendations/inputs, given in a structured manner, can help these schools have a greater stakeholder involvement, and create accountability for the teachers there.

Reviewer #2: The topic selected was well discussed area in Indian and international literature and there is no need to prove this hypothesis again and again.

It seems the authors have not mention about similar study findings in their literature survey.

There is a similar Indian study which has been done using National Family Health Survey data (Karnataka Health Promotion Trust (KHPT), Bangalore, Gouda M, S., & Sekher, Dr. T. V. (2014). Factors Leading to School Dropouts in India: An Analysis of National Family Health Survey-3 Data. IOSR Journal of Research & Method in Education (IOSRJRME), 4(6), 75–83. https://doi.org/10.9790/7388-04637583). The findings seems to be very much similar to the current article.

The data which were used were taken from a national data base and it appears that all the factors associated with school dropouts were not included. By just analysing the given factors and arriving at a conclusion is not really scientific.

Though the article has been written well and statistical methods were used accurately, the content is not worth publishing in a peer reviewed journal.

Reviewer #3: I appreciate the authors’ attention to strengthen the relationship between parental involvement in primary school and dropout in high school. This study fills up a research gap in the long-term consequence of parental involvement in children’s education, especially provides recommendations for the education policies in the local area. This study is a large-scale study based on two successive surveys 7 years apart, so the large-sampled size and the long following time are the advantages as a panel study. However, the authors wrote a little more on the value of the study and control variables effecting dropout and didn’t take advantage of the data of stratified analysis in the discussion part. I also suggest that more work in analyzing the data of parenting variables in the result part need to do and more literatures support that in the discussion to enrich the article. Meanwhile this article doesn’t provide ethical claim and information of data quality control as a large-size sampled study.

I have made some additional notes below in the event the authors wish to revise the article.

Abstract

The sentence “Additionally, the study provides evidence of……” (Line 38-40) should not presented in the introduction, and it should be integrated with the sentence “Further, a similar relationship was observed ……” (Line 48-49) in the results.

The results in the abstract need to present the data of stratified analysis.

Methods

The Round-I and Round-II data were all from India Human Development Survey, and the cited references are available in a website. However, the information of informed consent and quality control are hard to find from the website, so the authors should also present them in the manuscript.

There are three binary indicators for parental involvement, but there should be more definition of parental involvement, for example, how many times within a year or how long per involvement, et al.

Results

Line 238-242: the contents belong to “Data source”, and they are advised to be located after Line 135-141.

The inclusion standard of dropout should be presented in “Outcome variables”.

Figure 1 is suggested to be deleted.

Table 2 has the similar variables with Table 3 and should be merged to Table 3. It’s a long list of presented variables and this will make readers confused to compare the two tables.

In Table 3, class performance, grade repetition, type of school, household wealth and type of community et al are factors influencing school dropout besides parental involvement indicators, and the results show the importance of all the significant factors, rather than only parental involvement. So if the authors want to emphasize the of parental involvement is more important than other significant factors, more data need to support it.

Table 4-6 are suggested to be merged into one table for more readability.

More details about the stratified analysis should be provided, and rates of parental involvement in different gender, type of school and type of community are recommended to be presented.

Discussion

Line 370-393: it’s nonbusiness with the relationship of parental involvement and school dropout, and it seems to weaken the importance of parental involvement.

More literatures are suggested to be cited for discussing the results of stratified analysis and interpreting how parental involvement affects dropout in different gender, type of school and type of community.

Conclusion

The present results support the importance of parental involvement, and further conclusions, such as meaningful parental involvement (Line 426), should be supported by more data.

6. PLOS authors have the option to publish the peer review history of their article (what does this mean?). If published, this will include your full peer review and any attached files.

Reviewer #1: No

Reviewer #2: **Yes: **Mahesh Kumbukage

Reviewer #3: No

---

## [Author Response · Author response to Decision Letter 0]

3 Mar 2021

MANUSCRIPT NUMBER: PONE-D-20-34729

SUBJECT: Response to reviewers

Dear Dr. Srinivas Goli,

Thank you for giving us the opportunity of submitting an improved version of our manuscript titled “Does Lack of Parental Involvement Affect School Dropout among Indian Adolescents? Evidence from a Panel Study”, for publication in the PLOS One-Journal. We are also grateful to the three anonymous reviewers for their insightful comments and suggestions. We appreciate the time and effort that you and the reviewers have put forward to provide valuable feedback towards the improvement of our paper. Kindly note, that we have incorporated all of the changes that were suggested by the reviewers. The modifications have been highlighted in yellow within the revised manuscript. A final version of manuscript has also been provided. Please see below for a point-by-point response to the each of individual reviewer’s comments and suggestions.

Yours Sincerely,

Authors

 

EDITOR COMMENTS TO AUTHORS

Revised and resubmit according to reviewer comments. Additionally, also look at heterogeneous effects of the parental involvement by their educational standards and economic position. Parental involvement always not be in positive direction. Teacher and Parent meetings practice is mostly in private schools, thus there might be sample selection bias in the analyses. Kindly, examine this point as well. I will give my detailed comments once your respond to the reviewer comments. 

Response: Dear sir, I completely agree with your comment. We have done stratified analysis in the present study to examine the heterogeneity. Yes, Teacher and Parent meetings practice is mostly in private schools, thus there might be sample selection bias in the analyses. This was controlled in stratified analysis in table-4. 

REVIEWER COMMENTS TO AUTHORS

Reviewer #1: 

The study fills a gap in the current knowledge space. It will add to the existing database and inform policy makers. However, in order to make it more robust, the following are suggested:

There could be certain concerns about the cohort size, as the adolescents were only about 50% of the original children covered. Although that seems to have been addressed, some more explanation of whether that skews the results in any manner can be made.

Response: We agree with the reviewer than more explanation regarding factors associated with censoring of observation in the panel dataset is necessary. Accordingly, we have added the following explanation:

We observed that percentage difference of children by demographic, socio-economic and geographic characteristics was similar between the cross-sectional and panel datasets. Only percentage distribution by age of the children (in years) varied by more than 2% between the two datasets.

Another issue worth flagging is that in India, a child completes class ten by the age of 16 years, or 17 at most. Thus, there is a natural drop out as it were, around that age. If a larger number of children in the cohort are grouped towards the 17-18 years band, they would in any case drop out; either for work, or lack of higher secondary schools or colleges nearby. This could skew the results, and needs to be addressed. Besides, distance is a major factor in parents not sending their children for higher education. It will add to the study if this can be addressed.

Response: We completely agree with the reviewer’s assertion that there is a natural dropout at the 17-18 years age band. The same is observed from Table-2 of this paper, where children aged 11-12 in round-I faced higher dropout when they became 17-18 during round-II (in comparison to their counterparts in the 8-9 band during round-I). Accordingly, to adjust for this effect we have included age of the children (in years) as a control dummy in the logistic regression models. 

We also agree with the reviewer that distance is a major factor behind parents not sending their children to schools. However, IHDS has collected data of distance of commonly used local school from the community (selected). We found no way to understand whether all children in a community attended the same school. Therefore, to avoid ecological fallacy we did not include distance as an independent variable in our study. However, we have cited this as a limitation of our study (381-383):

The Right to Education Act in India addresses the entitlements of children up to 14 years. The authors could analyze whether this is a factor that comes into play in dropouts, as many benefits associated with the Act are not available after that age.

Response: We agree with the suggestion of reviewer. And accordingly, we have also mentioned the benefits of act on children during primary education, in our introduction section. Prior discussing how the dropout rate had increased during secondary schooling. However, there is no such data available in the survey which can show that this affects their dropout status. So, as per the reviewer’s suggestion we have cited this as a limitation.

The text in the limitation section reads as follows:

We were not able to capture the effect of Right to Education act entitled for under 15 years age children on their dropout status in later ages due to unavailability of data in survey.

The study has brought out the need for parental involvement in furthering the educational attainment of their children. Towards this end, some policy level suggestions can be incorporated. These could be structured parent teacher meetings, some activities for the family as a part of the “home work” etc. Even though the dropout rates from public schools are higher as brought out by the study, yet these recommendations/inputs, given in a structured manner, can help these schools have a greater stakeholder involvement, and create accountability for the teachers there.

Response: As per the reviewer’s suggestion, we have included this as our policy recommendation in the manuscript. The text reads as follows:

A structured implementation of policies which would help in holding parent-teacher meets, activities for the family as a part of homework, involvement of parents during child education etc are required to create a healthy environment among children-parents-teachers.

 

Reviewer #2: 

The topic selected was well discussed area in Indian and international literature and there is no need to prove this hypothesis again and again.

Response: Thank you for the suggestion. However, we respectfully disagree with the reviewer’s comment and would like to apologize if we were unable to convey our concept. We would like to mention that, although the concept of parental involvement in adolescent’s education have been widely shown in developed and developing countries. But such an idea is merely discussed in Indian context. The limited literature present shows a cross-sectional view of the scenario. The panel structure of our study itself adds the relevance of the paper. 

It seems the authors have not mention about similar study findings in their literature survey.

There is a similar Indian study which has been done using National Family Health Survey data (Karnataka Health Promotion Trust (KHPT), Bangalore, Gouda M, S., & Sekher, Dr. T. V. (2014). Factors Leading to School Dropouts in India: An Analysis of National Family Health Survey-3 Data. IOSR Journal of Research & Method in Education (IOSRJRME), 4(6), 75–83. https://doi.org/10.9790/7388-04637583). The findings seems to be very much similar to the current article.

Response: Thank you for the suggestion. However, we would respectfully like to mention that our manuscript has already discussed the above-mentioned paper written by Dr TV Sekher and M.S. Gouda. Although this literature using National Family Health Survey had helps us to provide various basic evidences, but as mentioned earlier it provides only a cross-sectional view of the scenario. And the present manuscript provides the long-term consequences of lack in parental involvement. You would be delighted to know we had the opportunity to discuss the current paper with Prof TV Sekher (we happen to work in the same institute). He had deemed the paper to address to crucial research gap.

The data which were used were taken from a national data base and it appears that all the factors associated with school dropouts were not included. By just analysing the given factors and arriving at a conclusion is not really scientific.

Response: Our study does not claim any causal effect but provides an idea how the parental involvement in primary level schooling is associated with the adolescent’s education outcome. And due to restriction in the information available in the data, all the factors might not able to be captured.

Though the article has been written well and statistical methods were used accurately, the content is not worth publishing in a peer reviewed journal.

Response: Dear sir, Thank you for the complement over our paper. I hope the changes made by us in the manuscript will make it worth publishing in a peer reviewed journal.

Reviewer #3: 

I appreciate the authors’ attention to strengthen the relationship between parental involvement in primary school and dropout in high school. This study fills up a research gap in the long-term consequence of parental involvement in children’s education, especially provides recommendations for the education policies in the local area. This study is a large-scale study based on two successive surveys 7 years apart, so the large-sampled size and the long following time are the advantages as a panel study. However, the authors wrote a little more on the value of the study and control variables effecting dropout and didn’t take advantage of the data of stratified analysis in the discussion part. I also suggest that more work in analyzing the data of parenting variables in the result part need to do and more literatures support that in the discussion to enrich the article. Meanwhile this article doesn’t provide ethical claim and information of data quality control as a large-size sampled study.

I have made some additional notes below in the event the authors wish to revise the article.

Abstract

The sentence “Additionally, the study provides evidence of……” (Line 38-40) should not presented in the introduction, and it should be integrated with the sentence “Further, a similar relationship was observed ……” (Line 48-49) in the results. The results in the abstract need to present the data of stratified analysis.

Response: Thank you for the suggestion. As per the reviewer’s suggestion, changes have been duly incorporated in the manuscript.

Methods

The Round-I and Round-II data were all from India Human Development Survey, and the cited references are available in a website. However, the information of informed consent and quality control are hard to find from the website, so the authors should also present them in the manuscript.

There are three binary indicators for parental involvement, but there should be more definition of parental involvement, for example, how many times within a year or how long per involvement, et al.

Response: We thank the reviewer for pointing out that the information on informed consent and data quality are hard to find from the website. Please note that the information on informed consent are available in the questionnaires of respective rounds of IHDS which can be downloaded along with the datasets (Desai et al., 2008; Desai & Vanneman, 2015). Further, information on data quality is separately available in the IHDS website (Desai et al., 2009). Furthermore, to accommodate your suggestion we have also modified the data availability information in the manuscript (line 130-134):

IHDS adopted a stratified random sampling survey design and informed consent were obtained from all the interviewee. Further details regarding survey description, sampling design and data quality can be found elsewhere (Sonalde Desai et al., 2009, 2010, 2015). Additional information on informed consent is available from the survey questionnaires available from the IHDS website (Sonalde Desai et al., 2008; Sonalde Desai & Vanneman, 2015).

Results

Line 238-242: the contents belong to “Data source”, and they are advised to be located after Line 135-141.

Response: According to the reviewer’s suggestion we have relocated the marked-up sentences (line 144-146).

The inclusion standard of dropout should be presented in “Outcome variables”.

Response: Thank you for pointing this out. We have given the inclusion criteria for school dropout as (line):

We have included only those children who were enrolled in a school during round-I.

Figure 1 is suggested to be deleted.

Response: According to the reviewer’s suggestion Figure 1 has been omitted.

Table 2 has the similar variables with Table 3 and should be merged to Table 3. It’s a long list of presented variables and this will make readers confused to compare the two tables.

Response: Thank you for the suggestion. Accordingly, we have merged Tables 2 and 3 into a single table named Table 2 below line 259.

In Table 3, class performance, grade repetition, type of school, household wealth and type of community et al are factors influencing school dropout besides parental involvement indicators, and the results show the importance of all the significant factors, rather than only parental involvement. So, if the authors want to emphasize the of parental involvement is more important than other significant factors, more data need to support it.

Response: Dear sir, we have only limited information available in dataset regarding the variables related to parental involvement. 

Table 4-6 are suggested to be merged into one table for more readability.

More details about the stratified analysis should be provided, and rates of parental involvement in different gender, type of school and type of community are recommended to be presented.

Response: According to the reviewer’s suggestion we have merged tables 4-6 into a single table. Further we have also provided the rates of parental involvement by gender, type of school and type of community in Table-3. In the present manuscript, more information is provided while explaining these tables. 

Discussion

Line 370-393: it’s non-business with the relationship of parental involvement and school dropout, and it seems to weaken the importance of parental involvement.

Response: Dear sir, I agree with your comment. But apart from parental involvement there are other factors to which are associated with school dropouts. Therefore, the first paragraph on the discussion was extensively discussed in accordance to the main objective of the paper and additionally, we tried to discuss other factors also which were significantly associated with school dropout among adolescents in India. 

More literatures are suggested to be cited for discussing the results of stratified analysis and interpreting how parental involvement affects dropout in different gender, type of school and type of community.

Response: Few literatures from India had so far shown differential in these stratified fields, however these were restricted to certain state only. These evidences were also discussed previously in our manuscript. Although we have included a sentence in the introduction section to brief the evidence.

Conclusion

The present results support the importance of parental involvement, and further conclusions, such as meaningful parental involvement (Line 426), should be supported by more data.

Response: Dear sir, we have only limited information available in dataset regarding the variables related to parental involvement.

References

Desai, S., Dubey, A., Joshi, B. L., Sen, M., Sharif, A., & Vanneman, R. (2009). India Human Development Survey: Design and Data Quality. University of Maryland and National Council of Applied Economic Research, New Delhi. https://www.icpsr.umich.edu/icpsrweb/content/DSDR/idhs-data-guide.html

Desai, S., & Vanneman, R. (2015). India Human Development Survey-II (IHDS-II), 2011-12: Version 6 [Data set]. Inter-University Consortium for Political and Social Research. https://doi.org/10.3886/ICPSR36151.V6

Desai, S., Vanneman, R., & National Council Of Applied Economic Research, New Delhi. (2008). India Human Development Survey (IHDS), 2005: Version 12 [Data set]. Inter-University Consortium for Political and Social Research. https://doi.org/10.3886/ICPSR22626.V12

---

## [Decision Letter · Decision Letter 1]

7 Apr 2021

PONE-D-20-34729R1

Does Lack of Parental Involvement Affect School Dropout among Indian Adolescents? Evidence from a Panel Study

PLOS ONE

Dear Dr. Srivastava,

Thank you for submitting your manuscript to PLOS ONE. After careful consideration, we feel that it has merit but does not fully meet PLOS ONE’s publication criteria as it currently stands. Therefore, we invite you to submit a revised version of the manuscript that addresses the points raised during the review process.

ACADEMIC EDITOR:

Considering reviewers opinion and my own reading, I am recommending a minor revision for this article. A part from reviewers comments, please address following comments from me:

1. In its current form, your discussion and conclusions looks mere superficial inferences without a proper understanding of Primary school education system, its characteristics and composition in India. I suggest to look at the following paper using the same data source which interpreted the characteristics of primary school education in India and how it is different by ownership types. Without understanding these dynamics, it is very difficult to suggest implications from your analyses. For instance, the special clause which is already mentioned in the existing flagship program like Sarva Siksha Abhiyan and right of children to free and compulsory education act, to strengthening community participation by forming village education committee (VEC) for the monitoring of the teachers’ activities as well as for the planning of infrastructure development. So, your suggestions should fit in existing structure for both government and private schools. Going by current composition, 70% of all school children are government where you don’t have proper set-up for parent and teacher interaction so one-way to involve them is through VEC system. You must try to understand the existing system for making policy recommendations. Please refer to below paper. 

Gouda J, Das KC, Goli S, Pou LM. Government versus private primary schools in India. International Journal of Sociology and Social Policy. 2013, Vol. 34(1/2), pp 708-724.

file://uniwa.uwa.edu.au/userhome/staff1/00102521/Downloads/GovernmentversusPrivateprimaryschoolsinIndia.pdf

We look forward to receiving your revised manuscript.

Kind regards,

Srinivas Goli, Ph.D.

Academic Editor

PLOS ONE

Journal Requirements:

Additional Editor Comments (if provided):

Considering reviewers opinion and my own reading, I am recommending a minor revision for this article. A part from reviewers comments, please address following comments from me.

1. In its current form, your discussion and conclusions looks mere superficial inferences without a proper understanding of Primary school education system, its characteristics and composition in India. I suggest to look at the following paper using the same data source which interpreted the characteristics of primary school education in India and how it is different by ownership types. Without understanding these dynamics, it is very difficult to suggest implications from your analyses. For instance, the special clause which is already mentioned in the existing flagship program like Sarva Siksha Abhiyan and right of children to free and compulsory education act, to strengthening community participation by forming village education committee (VEC) for the monitoring of the teachers’ activities as well as for the planning of infrastructure development. So, your suggestions should fit in existing structure for both government and private schools. Going by current composition, 70% of all school children are government where you don’t have proper set-up for parent and teacher interaction so one-way to involve them is through VEC system. You must try to understand the existing system for making policy recommendations. Please refer to below paper.

Gouda J, Das KC, Goli S, Pou LM. Government versus private primary schools in India. International Journal of Sociology and Social Policy. 2013, Vol. 34(1/2), pp 708-724.

file://uniwa.uwa.edu.au/userhome/staff1/00102521/Downloads/GovernmentversusPrivateprimaryschoolsinIndia.pdf

Reviewers' comments:

Reviewer's Responses to Questions

**Comments to the Author**

1. If the authors have adequately addressed your comments raised in a previous round of review and you feel that this manuscript is now acceptable for publication, you may indicate that here to bypass the “Comments to the Author” section, enter your conflict of interest statement in the “Confidential to Editor” section, and submit your "Accept" recommendation.

Reviewer #2: All comments have been addressed

Reviewer #3: (No Response)

2. Is the manuscript technically sound, and do the data support the conclusions?

Reviewer #2: Yes

Reviewer #3: Yes

3. Has the statistical analysis been performed appropriately and rigorously? 

Reviewer #2: Yes

Reviewer #3: Yes

4. Have the authors made all data underlying the findings in their manuscript fully available?

Reviewer #2: Yes

Reviewer #3: Yes

5. Is the manuscript presented in an intelligible fashion and written in standard English?

Reviewer #2: Yes

Reviewer #3: Yes

6. Review Comments to the Author

Reviewer #2: The earlier comments were addressed by the author. As the author has justified the reason for the publication, it is justifiable to publish this article

Reviewer #3: Thank your for your response, but still I have a concern. Since this paper focused on parental involvement, information of parenting should be addressed more. Otherwise, the title of this paper was only to attract attentions. Lack of further definition of parental involvement, such as how many times performed within a year or how long spent per involvement, should be presented as limitations.

7. PLOS authors have the option to publish the peer review history of their article (what does this mean?). If published, this will include your full peer review and any attached files.

Reviewer #2: **Yes: **Mahesh Kumbukage

Reviewer #3: No

---

## [Author Response · Author response to Decision Letter 1]

12 Apr 2021

Editor Comments

Considering reviewers opinion and my own reading, I am recommending a minor revision for this article. A part from reviewers’ comments, please address following comments from me.

1. In its current form, your discussion and conclusions looks mere superficial inferences without a proper understanding of Primary school education system, its characteristics and composition in India. I suggest to look at the following paper using the same data source which interpreted the characteristics of primary school education in India and how it is different by ownership types. Without understanding these dynamics, it is very difficult to suggest implications from your analyses. For instance, the special clause which is already mentioned in the existing flagship program like Sarva Siksha Abhiyan and right of children to free and compulsory education act, to strengthening community participation by forming village education committee (VEC) for the monitoring of the teachers’ activities as well as for the planning of infrastructure development. So, your suggestions should fit in existing structure for both government and private schools. Going by current composition, 70% of all school children are government where you don’t have proper set-up for parent and teacher interaction so one-way to involve them is through VEC system. You must try to understand the existing system for making policy recommendations. Please refer to below paper.

Gouda J, Das KC, Goli S, Pou LM. Government versus private primary schools in India. International Journal of Sociology and Social Policy. 2013, Vol. 34(1/2), pp 708-724.

file://uniwa.uwa.edu.au/userhome/staff1/00102521/Downloads/GovernmentversusPrivateprimaryschoolsinIndia.pdf

Response: We are thankful to receive your suggestions. Accordingly, we have incorporated changes in discussion and conclusion section (Page 18, Line 349; Page 20, Line 401). 

Reviewer #3: Thank you for your response, but still I have a concern. Since this paper focused on parental involvement, information of parenting should be addressed more. Otherwise, the title of this paper was only to attract attentions. Lack of further definition of parental involvement, such as how many times performed within a year or how long spent per involvement, should be presented as limitations.

Response: Dear Reviewer, Thank you for your suggestion. We have tried to use the best possible variable available in the data about the parent’s involvement in child learning process. We agree with your suggestion that quality variables like number of times these activities were performed and the time of involvement are the essential factors. But due to unavailability of these information in data, we have included this as the limitation. The text in the discussion section reads as follows (Page 19, Line 381):

Although factors like number of times and for how much time such involvement were observed also matter while observing the parental involvement. Unavailability of such information in the survey restricts the dimension.

---

## [Editor Report · Decision Letter 2]

19 Apr 2021

PONE-D-20-34729R2

Does Lack of Parental Involvement Affect School Dropout among Indian Adolescents? Evidence from a Panel Study

PLOS ONE

Dear Dr. Srivastava,

Thank you for submitting your manuscript to PLOS ONE. After careful consideration, we feel that it has merit but does not fully meet PLOS ONE’s publication criteria as it currently stands. Therefore, we invite you to submit a revised version of the manuscript that addresses the points raised during the review process.

ACADEMIC EDITOR: Before recommending this paper, I suggest authors to format their paper according to PLOS One guidelines. References are not in PLOS format. Give full details of newly cited reference. Convert all in text reference citation to numbers. Language need to be proof read once, especially for the newly added portions. PLOS one editorial central alerting me that this is a duplicate submission. Are you submitted the same paper in PLOS One or to the any other journal. 

We look forward to receiving your revised manuscript.

Kind regards,

Srinivas Goli, Ph.D.

Academic Editor

PLOS ONE

Journal Requirements:

Additional Editor Comments (if provided):

Before recommending this paper, I suggest authors to format their paper according to PLOS One guidelines. References are not in PLOS format. Give full details of newly cited reference. Convert all in text reference citation to numbers. Language need to be proof read once, especially for the newly added portions. PLOS one editorial central alerting me that this is a duplicate submission. Are you submitted the same paper in PLOS One or to the any other journal.

---

## [Author Response · Author response to Decision Letter 2]

25 Apr 2021

Response: Thank you for pointing this out. Indeed, we manually reviewed the reference list and checked for the presence of retracted articles. We found no retracted articles in the manuscript.

Additional Editor Comments:

1. Before recommending this paper, I suggest authors format their paper according to PLOS One guidelines. 

Response: Dear Editor, thank you for pointing this out. We have formatted the paper according to the PLOS One guidelines. Formatting changes have been shown in the “Revised manuscript with Track Changes”.

2. References are not in PLOS format. Give full details of the newly cited reference. Convert all in-text reference citation to numbers. 

Response: Dear Editor, thank you for pointing this out. We have modified the references according to the PLOS One format. Further to improve the veracity of our findings we included the following reference in the Discussion section:

Gouda J, Das KC, Goli S, Pou LMA. Government versus private primary schools in India. International Journal of Sociology and Social Policy. 2013.

Further, we have converted all in-text citations to numbers.

3. Language needs to be proofread once, especially for the newly added portions. 

Response: Thank you for pointing this out. Accordingly, we have proofread the final manuscript. All changes from proofreading have been shown in the “Revised manuscript with Track Changes”.

4. PLOS one editorial central alerting me that this is a duplicate submission. Have you submitted the same paper in PLOS One or to any other journal?

Response: We reconfirm that the same paper has neither been submitted to PLOS One nor any other journal. However, an older version of the paper was sent for the PAA Annual Seminar 2021.

---

## [Editor Report · Decision Letter 3]

28 Apr 2021

Does Lack of Parental Involvement Affect School Dropout among Indian Adolescents? Evidence from a Panel Study

PONE-D-20-34729R3

Dear Dr. Srivastava,

We’re pleased to inform you that your manuscript has been judged scientifically suitable for publication and will be formally accepted for publication once it meets all outstanding technical requirements.

Kind regards,

Srinivas Goli, Ph.D.

Academic Editor

PLOS ONE

Additional Editor Comments (optional):

Now, revisions are satisfactory and this paper can be accepted.
---

## [Editor Report · Acceptance letter]

30 Apr 2021

PONE-D-20-34729R3 

Does Lack of Parental Involvement Affect School Dropout among Indian Adolescents? Evidence from a Panel Study 

Dear Dr. Srivastava:

I'm pleased to inform you that your manuscript has been deemed suitable for publication in PLOS ONE. Congratulations! Your manuscript is now with our production department. 

Kind regards, 

on behalf of

Dr. Srinivas Goli 

Academic Editor

PLOS ONE